# Deformation Analysis of Continuous Milling of Inconel718 Nickel-Based Superalloy

**DOI:** 10.3390/mi13050683

**Published:** 2022-04-27

**Authors:** Xueguang Li, Yahui Wang, Liqin Miao, Wang Zhang

**Affiliations:** 1College of Mechanical and Electrical Engineering, Changchun University of Science and Technology, Changchun 130022, China; zhangwang5434@sina.com; 2Sinohydro Bureau 11 Co., Ltd., Zhengzhou 450000, China; 11wangyahui@sina.com; 3Department of Numerical Design &Manufacturing, Changchun Research Institute of Equipment and Technology, Changchun 130012, China; mlq_1980@sina.com

**Keywords:** milling force, milling temperature, multiple linear regression, range analysis, chip morphology, tool wear, surface quality

## Abstract

As a difficult-to-process material, Inconel718 nickel-based superalloy is more and more widely used in aerospace, ocean navigation, and large-scale machinery manufacturing. Based on ABAQUS simulation software, this paper takes the milling force and temperature in the milling process of the nickel-based superalloy as the research object, and establishes the empirical formula for the prediction model of cutting force and cutting temperature based on the method of multiple linear regression. The significance of the prediction model was verified by the residual analysis method. Through data analysis, it is obtained: within a certain experimental range, the influence degrees of each milling parameter on the cutting force and cutting temperature are $f_{z}>a_{p}>n$ and $f_{z}>a_{p}\approx n$, respectively. The actual orthogonal cutting test was carried out on the machine tool, and the reliability and accuracy of the prediction model of cutting force, cutting temperature and tool wear amount were verified. The model formulas of the shear velocity field, shear strain field and shear strain rate field of the main shear deformation zone are constructed by using mathematical analysis methods. The influence law of cutting speed and tool rake angle on the variables of main shear zone is calculated and analyzed. Through the combination of theory and experiment, the relationship between cutting force, chip shape and machined surface quality in milling process was analyzed. Finally, with the increase in the cutting force, the serration of the chip becomes more and more serious, and the roughness of the machined surface becomes greater and greater.

## 1. Introduction

High-temperature alloys, also known as heat-strength alloys, heat-resistant alloys or superalloy, are mainly based on iron, cobalt, and nickel; high-temperature alloys are those which can withstand complex stress and undergo high-temperature service in a very harsh environment. They are widely used in aviation, aerospace, ship, power and petrochemical industries, and are key materials in rocket engines and aviation jet engines [1,2]. Rare metals (molybdenum, niobium, tungsten, etc.) added to nickel-based superalloys give these materials high corrosion resistance and high temperature oxidation resistance. Furthermore, it increases the stability of the materials and creates hard spots inside the matrix, which makes the cutting efficiency of such materials low and causes severe tool wear. Therefore, using experimental methods to study cutting force and cutting heat is time-consuming, labor-intensive, and the cost is relatively high. At present, simulation processing and numerical simulation have been widely used in the research of cutting force, cutting temperature, chip morphology, tool wear, etc. [3,4,5].

Lu Xiaohong et al. [6] studied the thermal-mechanical coupling analysis of micro-milling of Inconel718 and proposed a modified three-dimensional analysis model of micro-milling force. In the model, the contact area is considered a moving finite-length heat source and an analytical cutting temperature model is proposed based on Fourier’s law. The coupling calculation of the micro-milling force model and the temperature model are carried out through an iterative process. Yao Yang, et al. [7] established an empirical formula between cutting force and cutting process parameters through the linear regression method and orthogonal experiment design scheme, and verified the accuracy and reliability of the model by variance analysis. Shafiul Islam et al. [8] established a mathematical model of axial cutting force and verified it through the designed experimental group. The experimental results are in good agreement with the theoretical simulation results. In addition, response surface methodology was employed to optimize the cutting parameters. Xiaobin Cui et al. [9] found that the serration of chips became more and more obvious as the cutting speed increased. During the formation process of the separated sawtooth, the high temperature in the shear band had a substantial effect on the initiation of the crack in the chip. When the cutting speed increased, the formation frequency of sawteeth increased with a decreasing growth rate and the tool chip contact length exhibited a decreasing trend.

Linjiang He et al. [10] found some very small sawteeth on the free surface of chips under low cutting speeds, which transformed into irregular serrated chips with increasing cutting speeds and into serrated chip with an adiabatic shear band under high cutting speeds.Fulin Wang et al. [11] found stress concentration appears and shear slipping occurs along the shear plane in the process of serrated chip formation. The strain rate on the shear slipping surface is much greater than other places and the temperature gradient perpendicular to the shear plane is relatively higher. Chen Ertao et al. [12] used ABAQUS finite element software to study the influence of tool rake angle on chip formation and its morphology, cutting force change, etc., and concluded that the chip size is related to the tool angle.

Seyed Mohammad Ebrahimi et al. [13] has found that for low feed rates, the chip form will be changed from continuous to sawtooth form. Wenlong Song et al. [14] found that cutting speed had a profound effect on the cutting temperature. However, in high-speed cutting, there was not much difference in cutting performance owing to the high temperature caused by the elevated cutting speed. Grzesik et al. [15] studied the tool wear in Inconel718 superalloy turning from the perspective of improving process performance and productivity, and concluded that the main wear of the tool is abrasive, and there is groove wear at the cutting edge. Hao Pengfei [16] mainly carried out the cutting test of nickel-based superalloy under the condition of high-pressure cooling, and discussed the corresponding wear mechanism of the tool under different wear morphologies according to the experimental results. At the same time, the influence of different cutting speeds on the tool wear profile was systematically analyzed using a scanning electron microscope and energy spectrum analyzer.Xie Liming, Cheng Ge et al. [17] first designed the orthogonal milling test plan, and then used the MATLAB analysis software to calculate the empirical formula of the Cr12MoV milling temperature prediction model. Finally, the order of the influence of cutting process parameters on milling temperature and the optimal combination of cutting factor levels are obtained by means of range analysis. Hao Zhaopeng, Cui Ruirui, Fan Yihang et al. [18] found that there is a cutting temperature that can minimize tool wear during the cutting process. The research results can provide a more convenient and effective method to select reasonable process parameters.

However, as far as the current research situation is concerned, overall research is relatively lacking, the research aspect is not very comprehensive to a certain extent, and there is still relatively little research on the impact of actual cutting tests on the cutting process. This paper uses multiple linear regression methods to establish the empirical formulas of the cutting force and cutting temperature prediction models, and verifies the accuracy of the prediction models through actual orthogonal cutting tests and range analysis methods to study the cutting mechanism of Inconel718 nickel-based superalloy. According to the cutting simulation model, the formation process of sawtooth chips in the cutting process of Inconel718 nickel-based superalloy is analyzed. Combined with actual cutting experiments, the general laws of chip shape, cutting force, machining surface quality and tool wear during the cutting process are analyzed, so as to provide a theoretical basis for the actual machining of superalloys.

## 2. Establishment of the Prediction Model of Cutting Force and Cutting Temperature

Research on cutting force and cutting temperature during machining is the basis for further research on cutting tools, workpiece material cutting performance, and tool wear. At the same time, cutting force and cutting temperature are also important physical quantities in machining and two of the key factors affecting the surface quality of the workpiece. In order to reasonably select the machining parameters during the high-speed cutting of Inconel718 nickel-based superalloy, it is necessary to effectively predict the cutting force and temperature during milling before machining.

In the actual machining process, the milling force and temperature are not only related to the milling parameters, but also affected by a series of factors such as tool wear and the clamping of the machining system. However, other factors are random and uncertain, so they cannot be summarized through experience or theory. In order to be able to obtain the prediction model formula of milling force and milling temperature, the following assumptions must be made about the machining process.

(1)In the milling process, it is necessary to ignore the influence of the machine tool itself, clamping conditions, workpiece shape, material, etc. on the cutting force and cutting temperature in the process, and set the machine tool as a rigid body without deformation.(2)The tool will inevitably wear out during the machining process. This factor will affect the various parameters of the tool, which will lead to changes in the cutting force and cutting temperature. In the modeling process of cutting force and cutting temperature, it is assumed that the tool is always free of wear, and the size of the cutting force and cutting temperature are not affected by tool wear.(3)It is assumed that the entire machining process has been orthogonal milling, ignoring the influence of possible milling angles on cutting force and cutting temperature.

Generally, J-C constitutive equations are used to simulate the behavior of workpiece materials during cutting [19]. Therefore, in this study, the J-C constitutive equation of plastic material, which includes the effect of temperature and strain rate strengthening, is used for establishing the workpiece model. This model can be expressed as follows:(1)$$\sigma =(A+B\varepsilon^{n})[1+c\ln\frac{\dot{\varepsilon }}{{\dot{\varepsilon }}_{0}}][1-(\frac{T-T_{r}}{T_{m}-T_{r}}{)}^{m}]\;(2-1)$$
where σ (MPa) is the flow stress; A (MPa) is the yield strength of the material under quasi-static conditions; B is the strain hardening coefficient of the material; n is the strain strengthening coefficient of the material; c is the strain rate sensitivity coefficient; m is the temperature sensitivity coefficient; T(K) is the material temperature; $T_{r}$(K) is the reference room temperature, K; $T_{m}$(K) is the melting temperature of the material; ${\dot{\varepsilon }}_{0}$ is the reference strain rate; ε is the strain of the material; $\dot{\varepsilon }$ is the strain rate. The J-C constitutive parameters of Inconel718 provided by Huichen and Xueren [20] are shown in Table 1.

### 2.1. Derivation of Empirical Formula

According to the principle of metal cutting, the index formula [21,22,23,24] for calculating cutting force and cutting temperature is:(2)$$\begin{array}{l} F=C_{F}\cdot a_{p}^{a_{1}}\cdot n^{a_{2}}\cdot f_{z}{}^{a_{3}}\\ T=C_{T}\cdot a_{p}^{b_{1}}\cdot n^{b_{2}}\cdot f_{z}{}^{b_{3}} \end{array}$$
where $C_{F}$ and $C_{T}$ are coefficients depending on the processing material and cutting conditions. The relationship among the feed speed $\nu_{f}$, the feed per tooth $f_{z}$, and the speed n can be expressed as:(3)$$\nu_{f}=f_{z}\cdot n\cdot z$$
where z is the number of milling cutter teeth, z=4.

Take the logarithm of both sides of Equation (1) to get:(4)$$\mathrm{lg}F=\mathrm{lg}C_{F}+a_{1}\cdot \mathrm{lg}a_{p}+a_{2}\cdot \mathrm{lg}n+a_{3}\cdot \mathrm{lg}f_{z}$$

Let $a_{0}=\mathrm{lg}C_{F},x_{1}=\mathrm{lg}a_{p},x_{2}=\mathrm{lg}n,x_{3}=\mathrm{lg}f_{z},y=\mathrm{lg}F$, we can get:(5)$$y=a_{0}+a_{1}\cdot x_{1}+a_{2}\cdot x_{2}+a_{3}\cdot x_{3}$$

This is a linear relationship between the dependent variable and the independent variable. $y_{i}$ is used to represent the experimental result; the independent variables are $x_{i1}$, $x_{i2}$, $x_{i3}$, and the experimental error is $\varepsilon_{i}$, where i=1∼16. Then, the multiple linear regression equation is as follows:(6)$$\left\{\begin{array}{c} y_{1}=\beta_{0}+\beta_{1}x_{11}+\beta_{2}x_{12}+\beta_{3}x_{13}+\varepsilon_{1}\\ y_{2}=\beta_{0}+\beta_{1}x_{21}+\beta_{2}x_{22}+\beta_{3}x_{23}+\varepsilon_{2}\\ \cdots \cdots \cdots \cdots \cdots \cdots \cdots \cdots \cdots \cdots \cdots \cdots \\ y_{i}=\beta_{0}+\beta_{1}x_{161}+\beta_{2}x_{162}+\beta_{3}x_{163}+\varepsilon_{16} \end{array}\right.$$

Expressed as a matrix:(7)Y=Xβ+ε

Among them:(8)$$\;\;\;Y\;\;\;=\left[\begin{array}{c} {\;\;\;y\;\;\;}_{1}\\ {\;\;\;y\;\;\;}_{2}\\ \;\;\;\ldots \;\;\;\\ {\;\;\;y\;\;\;}_{16} \end{array}\right]\;X=\left[\begin{array}{cccc} 1 & x_{11} & x_{12} & x_{13}\\ 1 & x_{21} & x_{22} & x_{23}\\ \ldots & \ldots & \ldots & \ldots \\ 1 & x_{161} & x_{162} & x_{163} \end{array}\right]\;\;\;\;\beta \;\;\;=\left[\begin{array}{c} {\;\;\;\beta \;\;\;}_{0}\\ {\;\;\;\beta \;\;\;}_{1}\\ {\;\;\;\beta \;\;\;}_{2}\\ {\;\;\;\beta \;\;\;}_{3} \end{array}\right]\;\;\;\;\varepsilon \;\;\;=\left[\begin{array}{c} {\;\;\;\varepsilon \;\;\;}_{1}\\ {\;\;\;\varepsilon \;\;\;}_{2}\\ \;\;\;\ldots \;\;\;\\ {\;\;\;\varepsilon \;\;\;}_{16} \end{array}\right]$$

Y, X, β, and ε are corresponding matrices. The least-square method is used to estimate the parameter β, and the regression coefficient $a_{0},a_{1},a_{2},a_{3}$ is the least square estimation of the parameter $\beta_{0},\beta_{1},\beta_{2},\beta_{3}$ (including the influence of ε); then, the regression equation can be expressed as:(9)$$\hat{y}=a_{0}+a_{1}x_{1}+a_{2}x_{2}+a_{3}x_{3}$$
where $\hat{y}$ is a statistical variable, where
(10)$$a=\left[\begin{array}{c} a_{0}\\ a_{1}\\ a_{2}\\ a_{3} \end{array}\right]$$

### 2.2. Orthogonal Simulation Experiment Design

In order to accurately calculate the empirical formula of the cutting force and cutting temperature prediction model, this article adopts the orthogonal experiment method. Orthogonal experimentation is a common method to study the influence of multiple factors. Compared with the comprehensive experiment, the orthogonal experiment selects some representative points from all combinations for experimenting. Through the experimental measurement values of these points, the influence of each factor on the results is analyzed. The advantage of the orthogonal experiment method is that the influence of cutting parameters on the object can be studied through fewer experiments.

In the research, the milling cutter is fixed, and it does not need to consider the influence of cutter parameters on the milling force. This paper designs 16 sets of orthogonal experiments with 3 factors and 4 levels. The experimental factors are axial cutting depth, spindle speed, and feed per tooth. Starting from reality, the value of each factor level is within the operating range of the machine tool. The range of experimental parameters is as follows:

(1)Spindle speed n: 3000∼4500 r/min;(2)Axial depth of cut $a_{p}$: 0.2∼0.5 mm;(3)Feed per tooth $f_{z}$: 0.1∼0.4 mm/r.

The orthogonal experiment and simulation data are shown in Table 2:

### 2.3. Data Processing and Determination of Model Coefficients

Taking the logarithm of the milling parameters in the table, the form of the matrix transformed by the multiple linear regression is:(11)$$Y_{F}=\left[\begin{array}{c} 2.2662\\ 2.5045\\ 2.8110\\ 2.8614\\ 2.6237\\ 2.5175\\ 2.8746\\ 2.8515\\ 2.8109\\ 2.8330\\ 2.4976\\ 2.7156\\ 2.6040\\ 2.7490\\ 2.7372\\ 2.5828 \end{array}\right]X=\left[\begin{array}{cccc} 1 & -0.6990 & 3.4771 & -1\\ 1 & -0.5229 & 3.4771 & -0.6990\\ 1 & -0.3979 & 3.4771 & -0.5229\\ 1 & -0.3010 & 3.4771 & -0.3979\\ 1 & -0.6990 & 3.5441 & -0.6990\\ 1 & -0.5229 & 3.5441 & -1\\ 1 & -0.3979 & 3.5441 & -0.3979\\ 1 & -0.3010 & 3.5441 & -0.5229\\ 1 & -0.6990 & 3.6021 & -0.5229\\ 1 & -0.5229 & 3.6021 & -0.3979\\ 1 & -0.3979 & 3.6021 & -1\\ 1 & -0.3010 & 3.6021 & -0.6990\\ 1 & -0.6990 & 3.6532 & -0.3979\\ 1 & -0.5229 & 3.6532 & -0.5229\\ 1 & -0.3979 & 3.6532 & -0.6990\\ 1 & -0.3010 & 3.6532 & -1 \end{array}\right]Y_{T}=\left[\begin{array}{c} 2.4663\\ 2.6251\\ 2.6933\\ 2.7749\\ 2.6114\\ 2.5942\\ 2.7821\\ 2.6889\\ 2.7344\\ 2.7895\\ 2.6065\\ 2.6701\\ 2.6257\\ 2.7715\\ 2.7480\\ 2.4576 \end{array}\right]$$

The M file is established by MATLAB software, and the cutting force value and each milling parameter in Formula (11) are calculated by multiple linear regression [25,26] to the numerical input program. The input program is:

y = [2.2662,2.5045,2.8110,2.8614,2.6237,2.5175,2.8746,2.8515,2.8109,2.8330,2.4976,2.7156,2.6040,2.7490,2.7372,2.5828];

x1 = [−0.6990,−0.5229,−0.3979,−0.3010,−0.6990,−0.5229,−0.3979,−0.3010,−0.6990,−0.5229,−0.3979,−0.3010,−0.6990,−0.5229,−0.3979,−0.3010];

x2 = [3.4711,3.4711,3.4711,3.4711,3.5441,3.5441,3.5441,3.5441,3.6021,3.6021,3.6021,3.6021,3.6532,3.6532,3.6532,3.6532];

x3 = [−1,−0.6990,−0.5229,−0.3979,−0.6990,−1,−0.3979,−0.5229,−0.5229,−0.3979,−1,−0.6990,−0.3979,−0.5229,−0.6990,−1]];

X = [ones(length(y),1),x1’,x2’ x3’];

Y = y’;

[b,bint,r,rint,stats] = regress(Y,X);

b,bint,stats

According to the results obtained by MATLAB:$$a_{0}=2.1253;\;a_{1}=0.4642;\;a_{2}=0.3260;\;a_{3}=0.5919.\;$$

Therefore, the multiple regression model is:$$y=2.1253+0.4642x_{1}+0.3260x_{2}+0.5919x_{3},\;$$
and the empirical formula of the cutting force prediction model available from above is:$$F=133.44\cdot a_{p}{}^{0.4642}\cdot n^{0.3260}\cdot f_{z}{}^{0.5919}.$$

Furthermore, take the logarithm of T in the table, as shown in Formula (11). Type in MATLAB:

y = [2.4663,2.6251,2.6933,2.7749,2.6114,2.5942,2.7821,2.6889,2.7344,2.7895,2.6065,2.6701,2.6257,2.7715,2.7480,2.4576];

According to the results obtained by MATLAB:$$b_{0}=2.5139;\;b_{1}=0.1275;\;b_{2}=0.1258;\;b_{3}=0.3610.\;$$

Therefore, the multiple regression model is:$$y=2.5139+0.1275x_{1}+0.1258x_{2}+0.3610x_{3},\;$$
and the empirical formula of the cutting force prediction model available from above is: $T=326.51\cdot a_{p}{}^{0.1275}\cdot n^{0.1258}\cdot f_{z}{}^{0.3610}$.

### 2.4. Significance Tests for Predictive Models

According to the results obtained by MATLAB:

(1)Cutting force regression coefficient group: $a_{0}=2.1253$, the confidence interval of $a_{0}$ is (−0.1699,4.4204), $a_{1}=0.4642$, the confidence interval of $a_{1}$ is (0.1722,0.7563), $a_{2}=0.3260$, the confidence interval of $a_{2}$ is (−0.3151,0.9670), $a_{3}=0.5919$, and the confidence interval of $a_{3}$ is (0.3998,0.7841). Statistics variable stats get: $r^{2}=0.8292$, F=19.4197, p=0.0001, obviously p<α=0.05. Perform residual analysis on the coefficient and enter in the MATLAB window: rcoplot(r,rint), the residual graph is shown in Figure 1a:

**Figure 1 micromachines-13-00683-f001:**
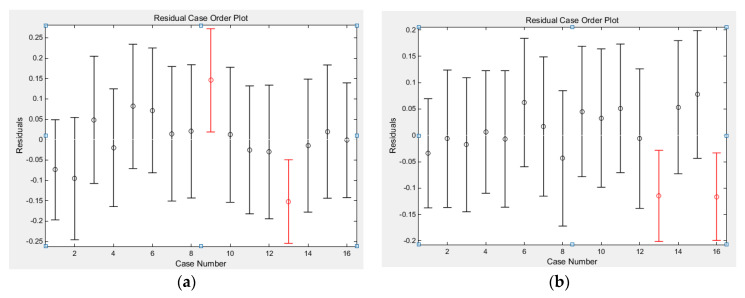
Residual analysis graph. (**a**) Analysis of residual cutting force. (**b**) Analysis of residual cutting temperature.

It can be seen from the residual analysis graph that, except for the 9th and 13th entries, the residuals of the remaining data are close to the zero point and the confidence interval of the residuals all contain the zero point, which shows that the regression model can better conform to the original data. The 9th and 13th data can be regarded as abnormal points, and the regression model is considered credible [27]. From the above, the regression model $y=2.1253+0.4642x_{1}+0.3260x_{2}+0.5919x_{3}$ is established, so the cutting force empirical formula $F=133.44\cdot a_{p}{}^{0.4642}\cdot n^{0.3260}\cdot f_{z}{}^{0.5919}$ is significant.

(2)Cutting temperature regression coefficient group: $b_{0}=2.5139$, the confidence interval of $b_{0}$ is (0.6733,4.3544), $b_{1}=0.1275$, the confidence interval of $b_{1}$ is (−0.1066,0.3617), $b_{2}=0.1258$, the confidence interval of $b_{2}$ is (−0.3883,0.6399), $b_{3}=0.3610$, and the confidence interval of $b_{3}$ is (0.2069,0.5152). Statistics variable stats get: $r^{2}=0.6981$, F=9.2480, p=0.0019, obviously p<α=0.05. Perform residual analysis on the coefficient and enter in the MATLAB window: rcoplot(r,rint), the residual graph is shown in Figure 1b.

It can be seen from the residual analysis graph that except for the 13th and 16th entries, the residuals of the remaining data are all close to the zero point, and the confidence interval of the residuals all contain the zero point, which shows that the regression model can better conform to the original data. The 13th and 16th data can be regarded as abnormal points, and the regression model is considered credible. From the above, the regression model $y=2.5139+0.1275x_{1}+0.1258x_{2}+0.3610x_{3}$ is established, so the cutting force empirical formula $T=326.51\cdot a_{p}{}^{0.1275}\cdot n^{0.1258}\cdot f_{z}{}^{0.3610}$ is significant.

### 2.5. Influence Rule of Cutting Process Parameters

The range analysis method can be used to obtain the primary and secondary order of the degree of influence of each factor on the target parameter, the optimal combination scheme, and the influence law of the influencing factor on the target parameter in the analysis of the results of the orthogonal experiment. In order to obtain the order of the influence degree of each factor on the milling force and milling temperature, the range analysis method is used to analyze the simulation results, as shown in Table 3 and Table 4:

Figure 2 is a trend graph showing the influence of spindle speed n, axial depth of cut $a_{p}$, and feed per tooth $f_{z}$ on cutting force F and cutting temperature T. Based on this, the changes in milling temperature can be seen when various parameters are changed. Analyzing the range table and indicator factor trend chart shows that:

(1)The degree of influence of each milling parameter on the cutting force (F) is $f_{z},a_{p},n$ in descending order, where $f_{z}$ has a larger degree of influence, $a_{p}$ has a medium influence on cutting force, and n has a smaller influence on cutting force. From the perspective of reducing the cutting force, the best combination of milling parameters is: $n=4000\;r/\min,a_{p}=0.4\;\mathrm{mm},f_{z}=0.1\;\mathrm{mm}/r$.(2)The degree of influence of each milling parameter on the cutting temperature (T) is $f_{z},a_{p},n$ in descending order, where $f_{z}$ has a larger degree of influence, and the influence of $a_{p}$ and n on cutting temperature is close. From the perspective of reducing the cutting temperature, the best combination of milling parameters is: $n=4500\;r/\min,a_{p}=0.5\;\mathrm{mm},f_{z}=0.1\;\mathrm{mm}/r$.(3)From Figure 2a,b, it can be seen that the milling force first increases and then decreases with the increase in the rotational speed. Since the shear angle increases with the increase in the rotational speed, the cutting deformation of the workpiece during the milling process gradually decreases, and the cutting force gradually decreases. When the rotation speed is lower than 4000 r/min, it can be seen that the change in cutting temperature shows an obvious upward trend, because during the machining process, severe friction occurs between the front and rear rake surfaces of the tool, the workpiece surface and the chip surface. These mutual frictions generate a lot of heat and gradually transfer it to other parts, so the temperature continues to increase. However, when the rotation speed continues to increase, the flow rate of the chips is also faster and more heat is taken away by the chip flow, so the temperature increases slowly.(4)It can be seen from Figure 2c,d that after the increase in the feed per tooth, the thickness of the material to be cut from the workpiece increases with each turn of the milling cutter, so the cutting force gradually increases. However, at the same time, after the feed per tooth increases, the cutting deformation coefficient of the workpiece material will decrease to a certain extent due to the influence of temperature, so the change in cutting force will gradually become gentle. When the feed per tooth is increased, more material is removed per revolution of the tool, so the tool will generate more and more heat when cutting the workpiece, and the cutting temperature changes significantly.(5)It can be seen from Figure 2e,f that with the gradual increase in the axial depth of cut, the area of the workpiece to be cut gradually increases, so the cutting force also increases. When the axial depth of cut is gradually increased, more heat needs to be generated to cut the workpiece material of the same path, so the change in cutting temperature will increase. However, in the actual machining process, as the axial depth of cut continues to increase, the contact length between the workpiece and the tool, the machining volume, etc. will increase significantly, and the machining environment at this time has changed significantly. Therefore, the cutting temperature does not increase in the same proportion.

## 3. Analysis of the Influence Law of Velocity Field, Strain Field and Strain Rate Field in the Main Shear Deformation Zone

According to the literature [28,29], the strain rate field $\dot{\gamma }$, strain field γ, and velocity field $\nu_{x}$ of the main shear deformation zone are as follows:$$\begin{array}{c} \dot{\gamma }=\left\{\begin{array}{c} \begin{array}{c} \frac{{\dot{\gamma }}_{m}}{{\left(\alpha h\right)}^{q}}y^{q}\;y\in [0,\alpha h] \end{array}\\ \begin{array}{c} \frac{{\dot{\gamma }}_{m}}{{\left(1-\alpha \right)}^{q}h^{q}}{\left(h-y\right)}^{q}\;y\in [\alpha h,h] \end{array} \end{array}\right.\\ \gamma =\left\{\begin{array}{c} \begin{array}{c} \frac{{\dot{\gamma }}_{m}}{(q+1)V\sin(\Phi ){\left(\alpha h\right)}^{q}}y^{q+1}\;y\in [0,\alpha h] \end{array}\\ \begin{array}{c} -\frac{{\dot{\gamma }}_{m}}{(q+1)V\sin(\Phi ){\left(1-\alpha \right)}^{q}h^{q}}{\left(h-y\right)}^{q+1}+\frac{\cos(\gamma_{o})}{\cos(\Phi -\gamma_{o})\sin(\Phi )}\;y\in [\alpha h,h] \end{array} \end{array}\right.\\ \nu_{x}=\left\{\begin{array}{c} \begin{array}{c} \frac{{\dot{\gamma }}_{m}}{(q+1){\left(\alpha h\right)}^{q}}y^{q+1}-V\cos(\Phi )\;y\in [0,\alpha h] \end{array}\\ \begin{array}{c} -\frac{{\dot{\gamma }}_{m}}{(q+1){\left(1-\alpha \right)}^{q}h^{q}}{\left(h-y\right)}^{q+1}+V\sin(\Phi )\tan(\Phi -\gamma_{o})\;y\in [\alpha h,h] \end{array} \end{array}\right. \end{array}$$
where α is the unequal coefficient, $\alpha =\frac{\cos\Phi \cos(\Phi -\gamma_{o})}{\cos(\gamma_{o})}$; h is the thickness of the main shear zone (mm), h = 0.025 mm; q is the exponent of the exponential function, which is 3 at low speed and 7 at high speed. V is cutting speed (m/min); Φ is shear angle (°); $\gamma_{o}$ is tool rake angle (°); ${\dot{\gamma }}_{m}$ is the maximum shear strain rate (s-1).

According to the above analysis, cutting speed and tool rake angle are the two main factors affecting the main shear deformation zone. The following will analyze the influence of these two factors on the main shear zone.

### 3.1. The Influence Law of Cutting Speed

When analyzing the influence law of cutting speed, the rake angle of the tool used is 12°, and the cutting speed range is 30∼120 m/min. The curves of the influence of the calculated cutting speed on the velocity field, shear strain field, and shear strain rate field in the main shear deformation zone are shown in Figure 3 and Figure 4.

It can be drawn from Figure 3a that: (1) In the main shear deformation zone, with the increase of the width of the main shear zone, the shear speed also increases. Furthermore, the size of the shear speed on the initial boundary (that is, the width of the main shear area on the figure is 0 mm) is affected by the initial cutting speed. (2) The tangential velocities on the final boundary (that is, the curves in the figure begin to stabilize) are all about 70 mm/s, and once the width of the main shear zone is greater than 0.02 mm, the change trend of the tangential velocity becomes gentle. (3) The main shear surface is defined as the plane where the tangential velocity component is equal to zero. It can be seen from Figure 3a that the lower the cutting speed is, the closer the location of the main shear surface (where the tangential velocity along the shear surface is 0) is to the starting boundary. That is to say, the lower the cutting speed, the smaller the width of the main cutting zone.
Figure 3The influence of cutting speed. (**a**) The influence of cutting speed on the tangential velocity along the shear plane. (**b**) The influence of cutting speed on shear strain.
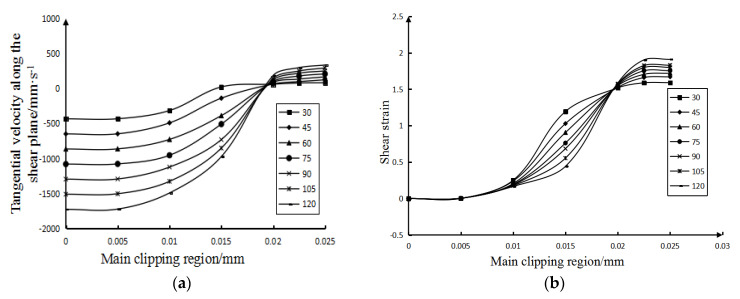


It can be concluded from Figure 3b: (1) Regardless of the cutting speed, the shear strain along the tangent direction of the shear plane at the beginning boundary of the main shear deformation zone is 0, because the material on the beginning boundary has not yet undergone any deformation. (2) Figure 3a shows the distance of the main shear zone corresponding to the main shear surface at each cutting speed. Therefore, it can be seen from Figure 3b that in the main shear deformation zone, the lower the cutting speed is, the closer the main shear surface is to the starting boundary. In fact, when the material crosses the final boundary of the main shear zone to form chips, its shear strain remains basically unchanged. Therefore, it can be seen that, the lower the cutting speed, the narrower the width of the main shear deformation zone. The analysis results are the same as above. (3) After crossing the main shear plane, the variation trend of the material shear strain gradually becomes gentle. This is because after the workpiece material crosses the main shear plane, the cut material forms chips and falls off the workpiece, and the tool no longer shears the chips, so the shear strain of the material tends to be stable.
Figure 4The influence of cutting speed on shear strain rate.
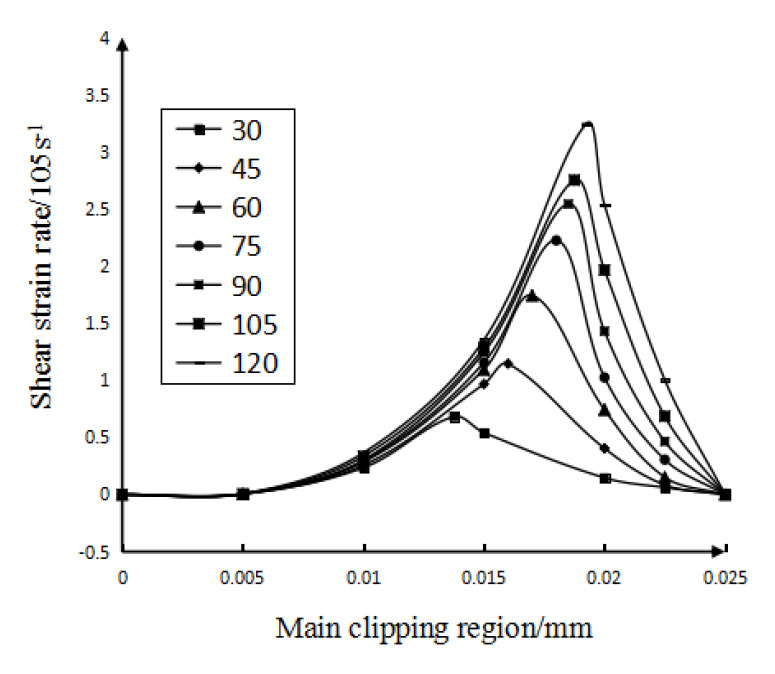


It can be concluded from Figure 4: (1) At the same cutting speed, with the gradual increase in the width of the main shear zone, the variation trend of the shear strain rate is to increase first and then decrease, and the greater the cutting speed, the greater the variation of the shear strain rate. (2) In the main shear zone, the shear strain rate on the main shear surface is the highest. It can also be seen from Figure 4 that the smaller the cutting speed, the closer the distance between the main shearing surface and the starting boundary. Furthermore, under each cutting speed condition, the shear surface in Figure 4 (the position with the highest shear strain rate) and the shear surface in Figure 3a are almost at the same position, which mutually confirms the correctness of the analysis results. (3) It can also be seen from Figure 4 that the lower the cutting speed, the smaller the maximum shear strain rate of the main shear deformation zone along the tangent direction of the shear plane; that is, the smaller the shear strain rate along the tangent direction of the main shear plane.

### 3.2. The Influence of Tool Rake Angle

When analyzing the influence law of the tool rake angle, the cutting speed adopted is 60 m/min and the range of tool rake angle is 0∼15°. Figure 5 and Figure 6 show the curves of the influence of the calculated tool rake angle on the velocity field, shear strain field, and shear strain rate field in the main shear deformation zone.

It can be seen from Figure 5: (1) At the same tool angle, the magnitude of the tangential velocity along the shear plane increases with the increase in the width of the main shear zone. However, at the initial boundary of the main shearing area (that is, where the width of the main shearing area is 0 mm in the figure), the size of the tool rake angle has little effect on the shearing speed. (2) On the final boundary (that is, as the curves in the figure begin to stabilize), the smaller the tool rake angle, the greater the tangential velocity along the shear plane. (3) It can be seen from the figure that the larger the degree of the rake angle of the tool, the farther the position of the main shear plane (that is, the tangential velocity along the shear plane in the figure is 0) is from the initial boundary. That is, the greater the degree of the tool rake angle, the greater the width of the main shearing area.

It can be seen from Figure 6a: (1) The rake angle of the tool has almost no effect on the shear strain of the starting boundary along the tangential direction of the shear plane. (2) The shear strain along the tangential direction of the shear surface at the final boundary of the main shear surface decreases with the increase in the rake angle of the tool. (3) When the tool rake angle is different, the position where the shear strain tends to be stable (where the curve tends to be stable) is not much different. It can be seen that the tool rake angle has little effect on the width of the main shear zone.
Figure 6The influence of tool rake angle. (**a**) The influence of tool rake angle on shear strain (**b**) The influence of tool rake angle on shear strain rate.
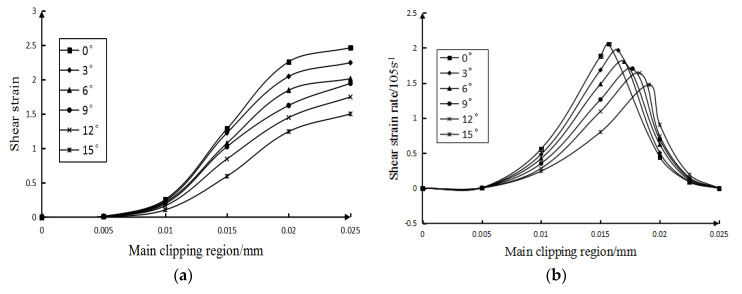


It can be seen from Figure 6b: (1) The larger the rake angle of the tool, the smaller the tangential shear strain rate of the main shear deformation zone along the shear plane. (2) On the basis of the analysis, under the condition of each tool rake angle, the shear surface in Figure 6b (where the position with the highest shear strain rate) and the shear surface in Figure 4 are almost at the same position. the rake angle of the tool does not have a great influence on the position of the cutting surface.

## 4. Analysis of High Temperature Alloy Milling Process

### 4.1. Cutting Superalloy Test

In order to verify the accuracy of the predicted value of the theoretical model, it is necessary to conduct cutting tests on superalloys. In this paper, a CNC machining center is used to conduct continuous milling experiments on Inconel718 nickel-based superalloy, and the cutting force, chip shape, tool wear profile and the quality of the machined surface are analyzed in the cutting process. After the experiment, a stereo microscope was used to detect tool wear and chip morphology, and a surface roughness measuring instrument was used to detect the machined surface.

Test machine

This experiment uses the H330 CNC machining center for cutting experiments, as shown in Figure 7a,b:

2.Cutting test materials

The milling cutter used was: YG8 hard alloy spiral mill tool, helix angle $\beta =45^{\circ }$, tool diameter D=10 mm, as shown in Figure 7c. The size of the test workpiece is: 100 mm × 100 mm × 20 mm, as shown in Figure 7d;

3.Experimental detection device

An Olympus SZ61 stereo microscope was used to observe the chip morphology. Its dimensions are 194 (W) × 253 (D) × 368 (H) mm, the handle stroke is 120 mm, the lens barrel tilt angle is 45° or 60°, the eyepiece is WHSZ15X-H FN 16*, and the objective lens is 110ALK0.4X 180–250. The NT1100 surface roughness measuring instrument is used to detect the surface quality of the processed workpiece. The longitudinal scanning range is 0.1−1 mm, the maximum scanning speed is 7.2 μm/s, and the sample stage size is 100 mm.The NT1100 surface roughness measuring instrument can perform fast, repeatable, and high-resolution three-dimensional measurement of the surface, and the measurement range can be from sub-nanometer roughness to millimeter step height.

### 4.2. Validation of the Prediction Model of Cutting Force and Cutting Temperature

Four groups of experiments were randomly conducted on the machining center to test the degree of agreement between the actual value of the data and the predicted value. Table 5 is as follows:

It can be seen from Table 5 that the average error of the milling force is about 6.04%, the milling temperature is about 7.14%, and the error between the actual measured value and the predicted value is relatively small, which shows that the predicted value of the theoretical formula is more consistent with the actual measured data. So, the theoretical prediction model can be used to predict the magnitude of force and temperature within the range of selected milling parameters.

### 4.3. Analysis of the Formation Process of Sawtooth Chips

In this section, one tip of the mill tool was taken as a research object to analyze the cutting process. The cutting parameters are simulated with a cutting speed of 18 m/min, a cutting depth of 0.4 mm, a tool rake angle of 12°, and a tool cutting edge radius of 0.01 mm to systematically study the formation process of sawtooth chips.

Figure 8 shows the change trend of the workpiece cutting temperature field with the cutting time at the initial stage of the formation of sawtooth chips. At t = 5 × 10^−4^ s, the tool has just contacted the workpiece, the highest temperature of the workpiece cutting temperature field is directly in front of the tool tip, then it decreases along the shear surface to the free surface of the workpiece, and the cutting temperature of the free surface of the workpiece is the lowest. At the same time, the area with the highest temperature is the smallest. Along the shear plane, the smaller the temperature, the larger the area.

During the cutting process, the temperature of the main shear zone is distributed in the above-mentioned manner. However, as the cutting time goes by, the maximum temperature of the workpiece cutting temperature field continues to rise. For example, when t = 5 × 10^−4^ s, it is 202 °C, when t = 8.8 × 10^−4^ s, it is 446 °C, when t = 1.2 × 10^−3^ s is 507 °C, and when t = 2.2 × 10^−3^ s is 599 °C. At the same time, the minimum temperature of the free surface of the workpiece also increases with the passage of cutting time, from t = 5 × 10^−4^ s to t = 2.2 × 10^−3^ s, and the temperature changes from 80 °C to 135 °C. From the above data analysis, it can be seen that the closer the shear surface is to the free surface of the workpiece, the slower the temperature increase rate is. In fact, when the temperature of the free surface of the workpiece reaches the softening temperature of the material, all materials on the shear surface have been softened, thus creating conditions for the formation of jagged chips. The softening temperature of the free surface material of the workpiece is the critical temperature for the formation of sawtooth chips. Based on the research in this article, the critical temperature for the formation of sawtooth chips of Inconel718 nickel-based superalloy is about 500 °C.
Figure 8Changes in the cutting temperature field at the beginning of cutting (°C). (**a**) t = 5 × 10^–^^4^ s (**b**) t = 8.8 × 10^–^^4^ s. (**c**) t = 1.2 × 10^–^^3^ s (**d**) t = 2.2 × 10^–^^3^ s.
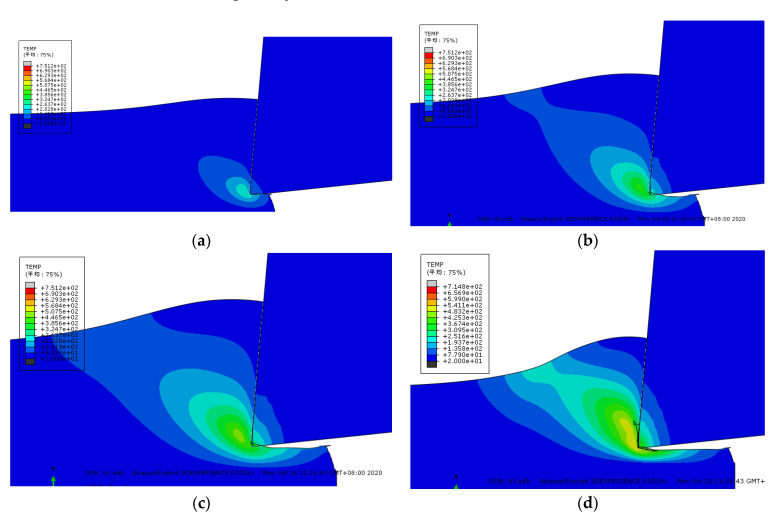


Figure 9 shows the simulation results of the formation of the second block of the sawtooth chip. When the tool moves to the moment of Figure 9a, the highest temperature of the cut area in front of the tool tip can reach 439 °C. However, the maximum temperature of the chip surface is above 668 °C, and the first segment of the saw-tooth chip is about to be completely formed. At the moment of Figure 9b, the first segment of the sawtooth chip has been completely formed, and the second segment has begun to form at the same time. At this time, the temperature in front of the tool tip reaches 501 °C. From the tool tip to the chip-free surface, the temperature decreases with the distance. As the cutting process progresses, the high temperature zone near the tool tip continuously spreads along the shear surface and covers the low-temperature zone near the chip-free surface. When the cutting proceeds to the moment in Figure 9c, the material in the first deformation zone continues to deform and the temperature continues to rise. In Figure 9e, the temperature of the material in the shear zone reaches about 682 °C. At this time, the thermal softening effect of the material is intensified, and the material on the shear surface undergoes concentrated slippage and plastic instability, so a convex surface is formed on the free surface of the workpiece. As the cutting progresses as shown in Figure 9f, the second block continues to slip due to the continued cutting of the tool. When proceeding to Figure 9g, the first deformation zone has been transferred to the second nodal block; so far, the second nodal block has been completely formed.

### 4.4. The Relationship between Cutting Force, Chip Morphology and Surface Quality

The actual cutting test is carried out to analyze the relationship between the cutting force, chip shape and the surface quality of the workpiece. By using Olympus SZ61 stereo microscope to observe the chip morphology, the NT1100 surface roughness measuring instrument is used to measure the surface quality after processing. As shown in Figure 10:

In this chapter, the influence of cutting factors is ignored when the relationship between cutting force, chip morphology and surface quality is studied, and only cutting force is used as the influencing factor. It can be seen from Figure 10a–h that the degree of chip sawtooth generated during the cutting process becomes larger and larger with the gradual increase in the cutting force, and the chip burr becomes sharper and sharper. In addition, the bottom part of the chip is relatively flat, and the other surface is sawtooth shape. It can be seen from Figure 10a that when the cutting force is low, the cutting speed at this time is relatively low, the shape of the chip is band-like, and the upper part of the chip has no clear saw teeth. However, when the machining speed increases, the serration of the chips becomes more and more serious, as shown in Figure 10c,e,g. This is because the shear force on the shear slip surface will increase sharply with the increase in the cutting speed, which will make the shear deformation on the shear slip surface more and more concentrated. Shear slip on the shear plane is also increasing, so chip serrations are also becoming more pronounced.

From Figure 10b, it can be seen that when the cutting force is low, the tool wear is not serious, so the processed surface quality is better at this time and the surface roughness is low. When the cutting force increases, as shown in Figure 10d,f,h, tool wear increases sharply, the degree of sawtooth chips deepens, and the scratches on the surface of the workpiece increase. At the same time, the increase in the cutting force increases the plastic deformation of the surface material of the workpiece, and the residual height of the processed surface increases, so the quality of the processed surface becomes worse and the surface roughness becomes larger and larger.

### 4.5. Research on the Wear Morphology of Cemented Carbide Tools

(1)Flank wear

Flank wear is the main wear form of carbide tools when milling high-temperature alloys. Flank wear is located at the tool flank near the main cutting edge. Flank wear is mainly caused by the friction between the tool’s flank and the rebounding workpiece surface, as shown in Figure 11a. The wear of the blade face is relatively uniform; dense and uniform wear streaks appear on the wear surface. From the characteristic point of view, the form of wear is abrasive wear. At the same time, it can be seen that the closer the tool tip, the greater the cutting force the tool is subjected to, and the more serious the wear that occurs. This wear feature belongs to a kind of boundary wear, which is also consistent with the previous conclusion.

(2)Wear of rake face

When milling high-temperature alloys, the rake face wear occurs, and the manifestation is crater wear, as shown in Figure 11b. The surface wear is located on the rake surface near the main cutting edge. It is mainly caused by the chemical interaction between the rake face of the tool and the high-temperature chips. In the milling process, the chips and the rake face are in direct contact and friction, so the chemical activity is very high. When the milling speed is relatively high, higher temperature and higher pressure will be generated, and it is easy to form crescent wear in this contact area. That is to say, the higher the cutting temperature, the more serious the tool wear.

(3)Layers of flaking

As shown in Figure 11c, the tool also has lamellar spalling during the milling process. This wear phenomenon is mainly reflected in the vicinity of the cutting edge of the tool, where the cutting force and cutting temperature are the largest. The main reason is that when the tool cuts into and out of the workpiece, contact fatigue and thermal stress are generated. Under the action of this stress, the surface of the tool edge gradually develops cracks until it falls off the surface of the tool in layers.

(4)Tool chipping

The chipping phenomenon of the tool mainly occurs at the tip of the tool, as shown in Figure 11d.As the tool continues to cut the workpiece, the work hardening of the workpiece is gradually strengthened, and the tool is continuously weakened. When the cutting force during the milling process is greater than the yield strength of the tool, the tip of the tool will be chipped. After a tool is chipped, the tool is “blunted,” and therefore, no longer capable of cutting the workpiece material.
Figure 11Wear morphology of tool. (*n* = 300 r/min, *a_p_* = 0.4 mm, *f_z_* = 0.4 mm/r). (**a**) Wear morphology of flank face. (**b**) Wear morphology of rake face. (**c**) Layered flaking. (**d**) Tool chipping.
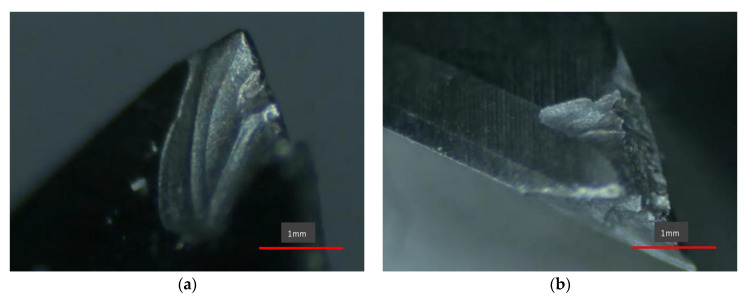

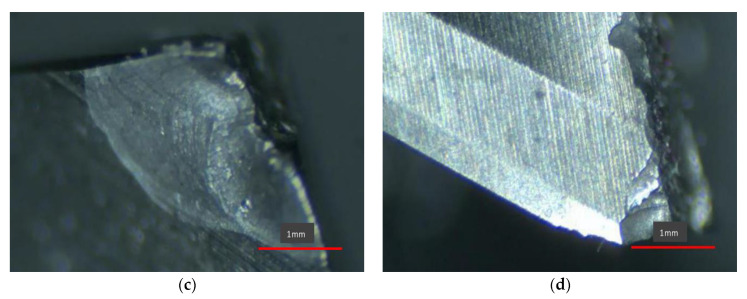


## 5. Conclusions

Based on the simulation analysis and actual experiment, this paper describes an Inconel718 nickel-base superalloy milling orthogonal experiment, establishes the cutting force and cutting temperature prediction model empirical formulas in the milling process, and obtains the following conclusions:

(1)Through orthogonal experiment analysis, the empirical formula of the cutting force prediction model is established as: $F=133.44\cdot a_{p}{}^{0.4642}\cdot n^{0.3260}\cdot f_{z}{}^{0.5919}$; the empirical formula of the cutting temperature prediction model is:$$T=326.51\cdot a_{p}{}^{0.1275}\cdot n^{0.1258}\cdot f_{z}{}^{0.3610}.$$(2)Through the range analysis of the result data, it is concluded that from the perspective of reducing the cutting force, the combination of the best milling parameters is: $n=4000\;r/\min,a_{p}=0.4\;\mathrm{mm},f_{z}=0.1\;\mathrm{mm}/r$, from the perspective of reducing the cutting temperature, the combination of the best milling parameters is: $n=4500\;r/\min,a_{p}=0.5\;\mathrm{mm},f_{z}=0.1\;\mathrm{mm}/r$.(3)The actual orthogonal cutting test was carried out on the machine tool to verify the reliability and accuracy of the prediction model of cutting force and cutting temperature.(4)With the velocity field, strain field and strain rate field model of the shear deformation zone, the influence curves of cutting speed and tool rake angle on shear speed, shear strain and shear strain rate are calculated and drawn. The influence of cutting speed and tool rake angle on shear speed, shear strain and shear strain rate are analyzed.(5)The mechanism of the formation of sawtooth chips is analyzed, and the distribution of stress, strain and temperature and the causes of formation are analyzed in detail by means of distribution clouds. Through a combination of theory and experiment, the relationship among cutting force, chip shape and surface quality in the milling process is analyzed. Finally, as the cutting force increases, the sawtooth of the chips becomes more and more serious, and the roughness of the machined surface becomes larger and larger.(6)The formation causes and formation processes of different tool wear morphologies are analyzed. In the high-efficiency milling of Inconel718 of cemented carbide end mills, the forms of tool wear are mainly blade spalling, tool chipping, tool surface pits and surface scratches. At the same time, tool front and flank wear will occur in high-speed cutting.

## Figures and Tables

**Figure 2 micromachines-13-00683-f002:**
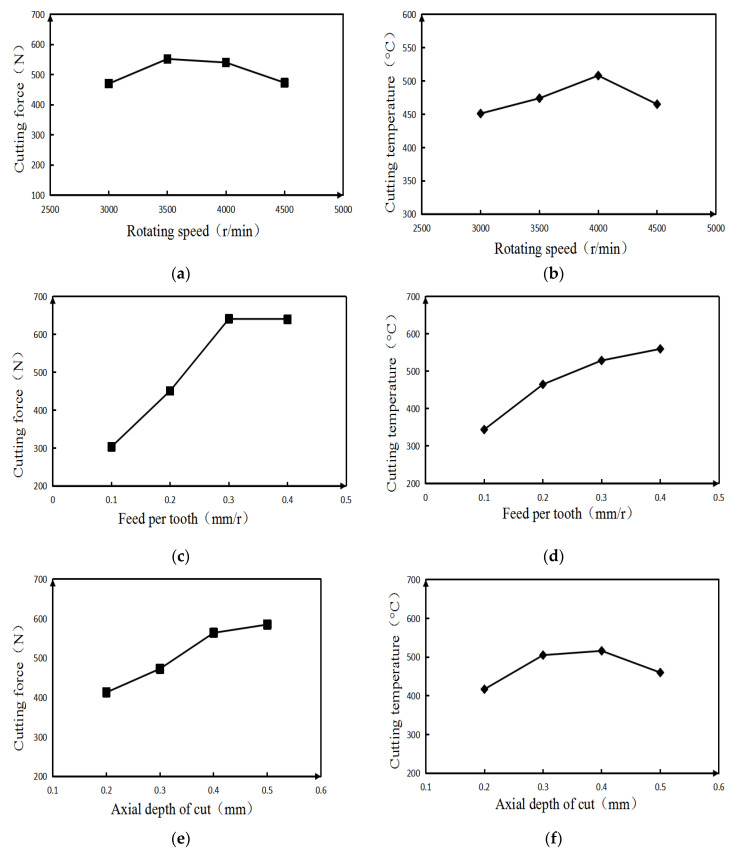
Trend chart of indicator factors. (**a**) The influence of spindle speed. (**b**) The influence of spindle speedon cutting forceon cutting temperature. (**c**) The influence of feed per tooth. (**d**) The influence of feed per tooth on cutting forceon cutting temperature. (**e**) Influence of axial depth of cut. (**f**) Influence of axial depth of cut on cutting forceon cutting temperature.

**Figure 5 micromachines-13-00683-f005:**
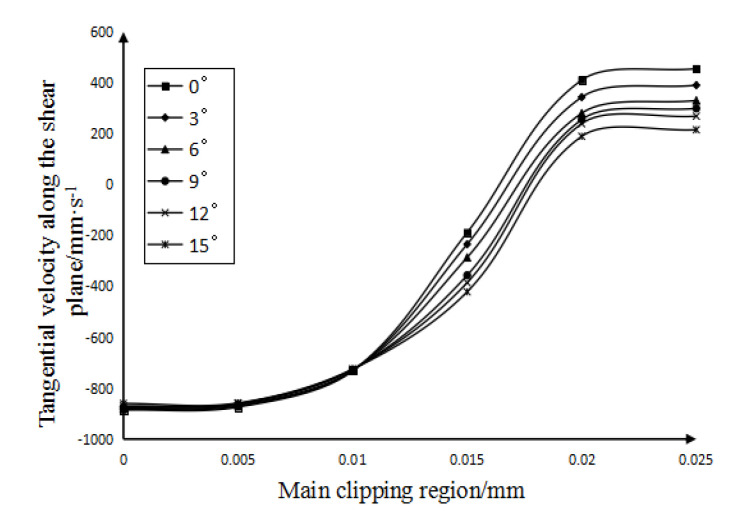
The influence of the tool rake angle on the tangential velocity along the shear plane.

**Figure 7 micromachines-13-00683-f007:**
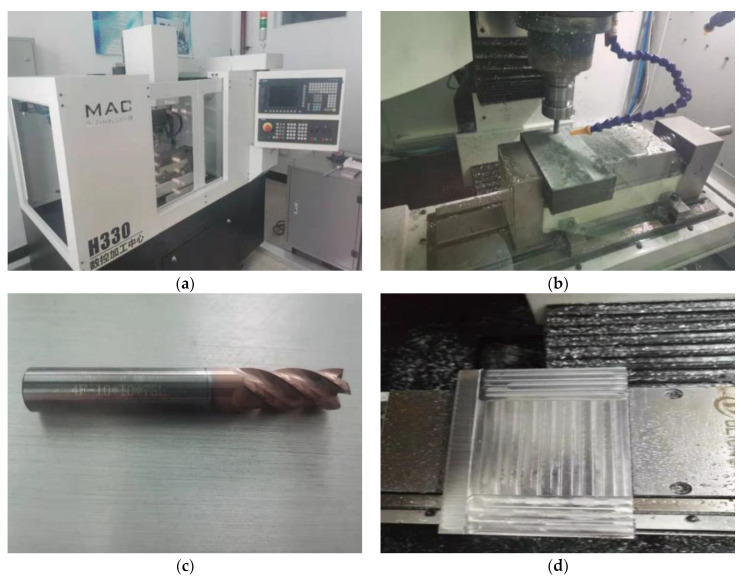
Experimental device. (**a**) CNC machining center. (**b**) The actual cutting test. (**c**) YG8 hard alloy spiral mill tool. (**d**) The finished part.

**Figure 9 micromachines-13-00683-f009:**
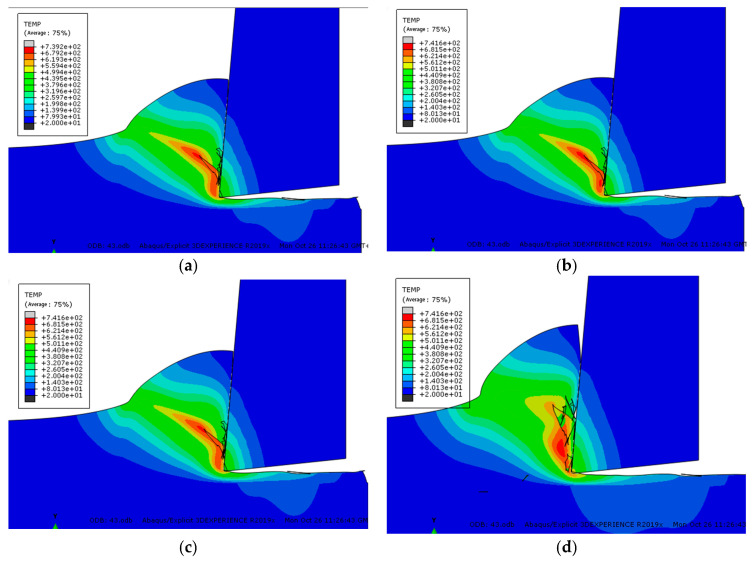
Changes in the cutting temperature field during the formation of serrated chips (°C). (**a**) The first section is aboutto be formed. (**b**) The first block isfully formed. (**c**) The second block begins to form. (**d**) The second nodal development process. (**e**) The second nodal mass starts to slip. (**f**) The second nodal slip process. (**g**) The second block is fully formed.

**Figure 10 micromachines-13-00683-f010:**
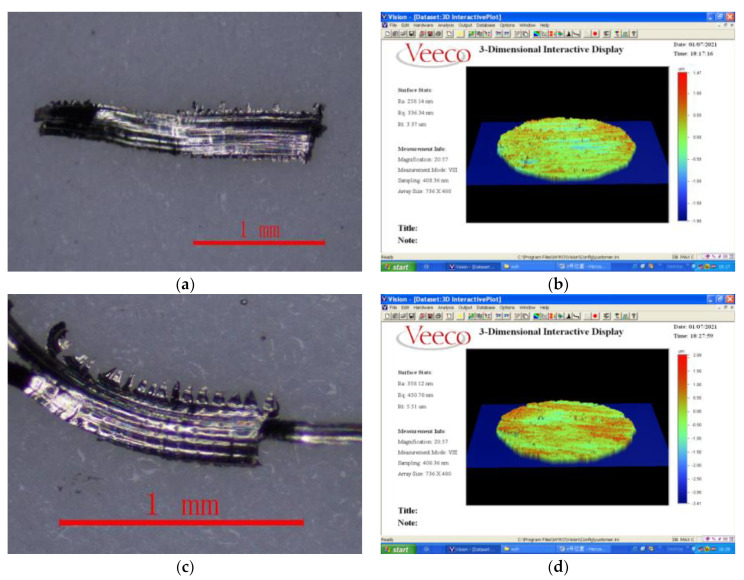
The shape of chips and roughness of surface. (**a**) Chip topography (Cutting force is 280 N). (**b**) Three-dimensional surface topography. (**c**) Chip topography (Cutting force is 400 N). (**d**) Three-dimensional surface topography. (**e**) Chip topography (Cutting force is 470 N). (**f**) Three-dimensional surface topography. (**g**) Chip topography (Cutting force is 620 N). (**h**) Three-dimensional surface topography.

**Table 1 micromachines-13-00683-t001:** Johnson-Cook plasticity parameters of Inconel718.

*A*	450 MPa
*B*	1700
$C({\dot{\varepsilon }}_{0}$ = 0.001)	0.017
n	0.65
*m*	1.3
Melting temperature	1570 °C
Excessive temperature	20 °C

**Table 2 micromachines-13-00683-t002:** Orthogonal simulation experiment data.

Numbering	Rotating Speed n(r/min)	Axial Depth of Cut $a_{p}(\mathrm{mm})$	Feed per Tooth $f_{z}(\mathrm{mm}/r)$	F (N)	T (°C)
1	3000	0.2	0.1	184.58	292.61
2	3000	0.3	0.2	319.55	421.76
3	3000	0.4	0.3	647.19	493.56
4	3000	0.5	0.4	726.77	595.48
5	3500	0.2	0.2	420.43	408.73
6	3500	0.3	0.1	329.22	392.81
7	3500	0.4	0.4	749.23	605.54
8	3500	0.5	0.3	710.36	488.57
9	4000	0.2	0.3	646.95	542.46
10	4000	0.3	0.4	680.82	615.83
11	4000	0.4	0.1	314.51	404.08
12	4000	0.5	0.2	519.54	467.89
13	4500	0.2	0.4	401.83	422.37
14	4500	0.3	0.3	561.11	590.91
15	4500	0.4	0.2	545.96	559.71
16	4500	0.5	0.1	382.66	286.79

**Table 3 micromachines-13-00683-t003:** Cutting force (F) range analysis table.

Level	Rotating Speed n(r/min)	Axial Depth of Cut $a_{p}(\mathrm{mm})$	Feed per Tooth $f_{z}(\mathrm{mm}/r)$
$K_{1}$	1878.09	1653.79	1210.97
$K_{2}$	2209.24	1890.70	1805.48
$K_{3}$	2161.82	2256.89	2565.61
$K_{4}$	1891.56	2339.33	2558.65
$k_{1}$	469.52	413.45	302.74
$k_{2}$	552.31	472.68	451.37
$k_{3}$	540.46	564.22	641.40
$k_{4}$	472.89	584.83	639.66
$R_{F}$	82.79	171.38	338.66
Primary and secondary order	$f_{z}>a_{p}>n$		

**Table 4 micromachines-13-00683-t004:** Cutting temperature (T) range analysis table.

Level	Rotating Speed n(r/min)	Axial Depth of Cut $a_{p}(\mathrm{mm})$	Feed per Tooth $f_{z}(\mathrm{mm}/r)$
$K_{1}$	1803.49	1666.17	1376.29
$K_{2}$	1895.65	2021.31	1858.09
$K_{3}$	2030.26	2062.89	2115.50
$K_{4}$	1859.78	1838.73	2239.22
$k_{1}$	450.87	416.54	344.07
$k_{2}$	473.91	505.33	464.52
$k_{3}$	507.57	515.72	528.88
$k_{4}$	464.95	459.68	559.81
$R_{T}$	56.7	99.18	215.74
Primary and secondary order	$f_{z}>a_{p}>n$		

**Table 5 micromachines-13-00683-t005:** Comparison of predicted and measured values.

Test Number	Rotating Speed (r/min)	Axial Depth of Cut (mm)	Feed per Tooth (mm/r)	*F* (N)			*T* (°C)		
				Predictive Value	Measured Value	Error (%)	Predictive Value	Measured Value	Error (%)
1	3000	0.4	0.4	689.55	730.00	5.95	571.39	533.79	6.58
2	3500	0.5	0.1	354.01	375.89	6.18	363.38	388.96	7.04
3	4000	0.3	0.3	558.91	526.55	5.79	514.77	554.61	7.74
4	4500	0.2	0.2	378.49	402.15	6.25	428.57	459.34	7.18
Average Error (%)				6.04			7.14		

## Data Availability

Not applicable.

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
