# Peer review of "Deformation Analysis of Continuous Milling of Inconel718 Nickel-Based Superalloy"

_micromachines, 2022, doi:10.3390/mi13050683_

Round 1

Reviewer 1 Report

The comments are as follows:

1.The article lacks innovation, but only establishes a prediction model by performing multiple linear fitting on the previous empirical formula, which has no essential innovation.

2.In general, the logic of the article is rather confusing.

3.The introduction part does not point out the shortcomings of the current research status, in order to highlight the innovation of the article.

4.The introduction of the orthogonal experiment principle in Section 2.2 is unnecessary.

5.Table 1 Missing units of T and F.

6.What does the red line segment in figure1 mean? And each figure in figure 2 has only one physical quantity, so the legend should be removed.

7.What is the constitutive model used by the simulation model, and the article lacks some details on the simulation model setup.

8.Most of the pictures in the article are blurry.

9.In the tool wear study in Section 4.4, it should be indicated under which conditions each wear occurs, and the analysis description is better linked to the previous work.

10.There are no pictures comparing the model prediction results with the experimental results.

11.Figure11 is missing a scale bar.

Author Response

  1. The article lacks innovation, but only establishes a prediction model by performing multiple linear fitting on the previous empirical formula, which has no essential innovation.

Reply: The innovation of this paper is that an orthogonal simulation experiment is established based on the ABAQUS simulation software, and the relationship formula between the process parameters, cutting force and cutting temperature is established by the method of multiple linear regression according to the data obtained from the experiment.The significance of the prediction model is verified by the residual analysis method, and the comparison process between the actual value and the predicted value has been added in the text.

  1. In general, the logic of the article is rather confusing.

Reply: The order of the relevant content in the article has been adjusted, and the entire article is studying the influence of various process parameters on cutting force and cutting temperature, and the logic confusion has been revised.

  1. The introduction part does not point out the shortcomings of the current research status, in order to highlight the innovation of the article.

Reply: The paper has added the shortcomings of the current research status.

  1. The introduction of the orthogonal experiment principle in Section 2.2 is unnecessary.

Reply: The introduction to the principle of orthogonal experiments has been deleted from the text.

  1. Table 1 Missing units of T and F.

Reply: The unit has been marked in the text.

  1. What does the red line segment in figure1 mean? And each figure in figure 2 has only one physical quantity, so the legend should be removed.

Reply: The red line in Figure 1 represents the data that does not contain zero points in the simulation experiment, and these two data can be regarded as abnormal data points.The legend in Figure 2 has been removed.

  1. What is the constitutive model used by the simulation model, and the article lacks some details on the simulation model setup.

Reply: This paper adopts the J-C constitutive model, and the relevant parameter settings of the constitutive model have been added to the text.

  1. Most of the pictures in the article are blurry.

Reply: Image clarity has been adjusted.

  1. In the tool wear study in Section 4.4, it should be indicated under which conditions each wear occurs, and the analysis description is better linked to the previous work.

Reply: The research on tool wear in this paper is mainly aimed at the wear profile of the tool during the cutting process. Therefore, the same working condition is selected as the wear occurrence condition, which has been indicated in the text, and the connection with the previous text has also been added.

  1. There are no pictures comparing the model prediction results with the experimental results.

Reply: Validation of prediction models with actual cutting tests has been added, and related analyses have been added.

  1. Figure11 is missing a scale bar.

Reply: The relevant pictures in the text have been modified.

Reviewer 2 Report

The manuscript can be accepted in its current format.

Author Response

  1. Moderate English changes required

Reply: All the manuscript have been checked and revised again.

Reviewer 3 Report

The title needs fine tuning to reflect better the content of the manuscript.

The abstract needs significant attention, as it does not summarise the work done and report the main findings.

The introduction has to be extended and more related work has to be included. Then, the gap has to be highlighted and followed with a clear aim and innovative contribution of this paper.

Section 2. Establishment of cutting force and cutting temperature prediction model, needs revision to improve its readability as it is difficult to follow in its current form.

Please rationalise the selection of the applied process parameters and their levels.

Where is the comparison between the results of the theoretical model and the experimental results, which should follow Section 4. 

The formation of serrated saw-tooth chip is not clear in the FE model, please adopt the model to obtain better results.

 Fig. 10 that supposed to show the relationship between generated surface and chip does not give sufficient quantitative information, it is just a visual unclear illustration. Please enhance.

How tool wear was characterised. Tool wear section suddenly appears at the end of the paper without clear link to the work done and presented. How tool wear affect and is considered in the developed model?

The conclusion needs revision to give insights into generic findings out of this research work.  

Author Response

  1. The title needs fine tuning to reflect better the content of the manuscript.

Reply: The title has been changed to Inconel 718 Nickel-Based Superalloy Milling Deformation Analysis.

  1. The abstract needs significant attention, as it does not summarise the work

done and report the main findings.

Reply: The abstract part of the text has been revised.

  1. The introduction has to be extended and more related work has to be included. Then, the gap has to be highlighted and followed with a clear aim and innovative contribution of this paper.

Reply: The introduction section has been revised.

  1. Section 2. Establishment of cutting force and cutting temperature prediction model, needs revision to improve its readability as it is difficult to follow in its current form.

Reply: The part of the prediction model of cutting force and cutting temperature has been added to the part of actual cutting test verification, which supplements the theory to a certain extent.

  1. Please rationalise the selection of the applied process parameters and their levels.

Reply: This paper mainly studies the change of cutting force and cutting temperature in the medium and high speed range, so the process parameters and range in this paper are selected.

  1. Where is the comparison between the results of the theoretical model and the experimental results, which should follow Section 4. 

Reply: A comparison of theoretical model results with experimental results has been added in Section 4.

  1. The formation of serrated saw-tooth chip is not clear in the FE model, please adopt the model to obtain better results.

Reply: In the milling process, the cutting that each cutting edge of the milling cutter participates in can be represented by two-dimensional cutting. Since the three-dimensional model is more complex and changeable, two-dimensional cutting can be used to reflect the general law of the main shear deformation zone of three-dimensional milling.

  1.  Fig. 10 that supposed to show the relationship between generated surface and chip does not give sufficient quantitative information, it is just a visual unclear illustration. Please enhance.

Reply: A quantitative relationship has been added to the relationship between surface quality and chip formation, and the relationship between the corresponding laws has been further explained.

  1. How tool wear was characterised. Tool wear section suddenly appears at the end of the paper without clear link to the work done and presented. How tool wear affect and is considered in the developed model?

Reply: The degree of tool wear is mainly related to the cutting force and cutting temperature, and the relevant description has been added in the text.

  1. The conclusion needs revision to give insights into generic findings out of this research work.  

Reply: The conclusions have been further revised.

Round 2

Reviewer 1 Report

The authors have addressed the reviewer comments and the manuscript was improved, so I suggest the paper publishing in Journal. Thanks to authors for their hard work.

Author Response

Dear reviewer, thank you so much for your support.

Reviewer 3 Report

The paper is difficult to follow and difficult to seethe contribution. The authors responded to the reviewer comments claiming that there they were addressed but unfortunately a couple of sentences added here and there.

The title, abstract and conclusion still need intensive revision.

It is difficult to understand how the claimed regression model was developed.

The accuracy of the results of the FE is questionable.

The selection of the applied process parameters is not rationalised.

This clearly indicates that the paper lacks novelty and accordingly I would suggest that the authors consider reworking the paper for a future submission with a focused problem statement and work to probe novel research findings that would help for an acceptable paper. It is unfortunate that I have to be negative but hopefully the comments will be interpreted constructively.

Author Response

Dear Reviewer,

Thank you for your advice and comments concerning our manuscript entitled “Analysis of Milling Deformation of Inconel718 Nickel-Based Superalloy”, those comments are all valuable and very helpful for revising and improving our paper, as well as the important guiding significance to our research. we have revised this manuscript and especially paid much attention to your comments and suggestions. We would like to re-submit it to micromachines and we sincerely hope it could meet with approval.

Best regards,

All the authors

Responses to Reviewer:

(1) The paper is difficult to follow and difficult to seethe contribution. The authors responded to the reviewer comments claiming that there they were addressed but unfortunately a couple of sentences added here and there.The title, abstract and conclusion still need intensive revision.

Reply: Dear reviewer, the Title has been changed to “Deformation analysis of continuous milling of Inconel718 nickel-based superalloy”, Abstract and Conclusions have been changed accordingly also.

(2) It is difficult to understand how the claimed regression model was developed.

Reply: Dear reviewer, the regression model is obtained based on the previous theoretical experience and combined with actual cutting processing tests. It is theoretically the same as other theoretical models, but it is different in terms of process parameters, cutting methods and cutting materials.

(3) The accuracy of the results of the FE is questionable.

Reply: Dear reviewer, all the finite element analysis parameters are set according to the actual working condition, the index of the prediction model is obtained by using Matlab software, which has certain feasibility and accuracy. The change curve of cutting force and cutting temperature is drawn according to the cutting data extracted from Abaqus software. The pictures and data of the actual cutting are also obtained through the actual cutting test and measured by cutting force tester and micro-thermal detector. The picture of tool wear is obtained with a stereo microscope and a grating surface profiler.

(4) The selection of the applied process parameters is not rationalised.

Reply: Dear reviewer, the process parameters are arranged based on actual cutting experiments. In this paper, the research on nickel-based superalloys is carried out in the environment of medium and high speed, so the rotation speed is selected in the range of medium and high speed, The research on cutting force and cutting temperature is to reflect the microscopic through the macroscopic test, so the axial depth of cut and the feed per tooth are more appropriate to choose within this range. The selection of process parameters within this range also clearly reflects the general law, but it cannot be fully reflected, so there will be certain defects and limitations.

(5) This clearly indicates that the paper lacks novelty and accordingly I would suggest that the authors consider reworking the paper for a future submission with a focused problem statement and work to probe novel research findings that would help for an acceptable paper. It is unfortunate that I have to be negative but hopefully the comments will be interpreted constructively.

Reply: Dear reviewer, in this paper, the influence law of cutting process parameters on cutting force and cutting temperature, the influence law on tool wear, the influence law on shear velocity field, shear strain field and shear strain rate field in the main shear deformation zone are mainly studied. The purpose is to study the deformation law of Inconel718 nickel-based superalloy during continuous milling. The main conclusions are as follows: Through the orthogonal test analysis, the empirical formula of the cutting force and cutting temperature prediction model is established. And through the actual orthogonal cutting test on the machine tool, the reliability and accuracy of the prediction model of cutting force and cutting temperature are verified. Finally, by analyzing the data, the optimal combination of milling parameters to reduce the milling force and milling temperature is obtained; Through the combination of theory and experiment, the relationship between cutting force, chip shape and surface quality during milling was analyzed. Finally, it is concluded that with the increase of cutting force, the sawtooth of the chip becomes more and more serious, and the roughness of the machined surface becomes larger and larger; By analyzing the causes and formation process of different tool wear patterns, it is concluded that the main forms of tool wear in milling are blade spalling, tool chipping, tool surface pits and surface scratches.

Thank you so much for your valuable suggestions, your comments are very important for us to go on the research further, and we will continue to deepen and improve our job in this research field in the coming future.
